# COVID-19: A complex disease with a unique metabolic signature

**Veronica Ghini[1,2], Walter Vieri[1,3], Tommaso Celli[4], Valentina Pecchioli[1,2], Nunzia Boccia[4], Tania Alonso-Vásquez[3], Lorenzo Pelagatti[4], Marco Fondi[3], Claudio Luchinat[1,2,5], Laura Bertini[4], Vieri Vannucchi[4], Giancarlo Landini[4]\*, Paola Turano [1,2,5]\***

**1** Department of Chemistry "Ugo Schiff", University of Florence, Sesto Fiorentino Florence, Italy, **2** Magnetic Resonance Center (CERM), University of Florence, Sesto Fiorentino, Florence, Italy, **3** Department of Biology, University of Florence, Sesto Fiorentino, Florence, Italy, **4** Internal Medicine, Santa Maria Nuova Hospital, Florence, Florence, Italy, **5** Consorzio Interuniversitario Risonanze Magnetiche di Metallo Proteine (CIRMMP), Sesto Fiorentino Florence, Italy

\* giancarlo.landini@uslcentro.toscana.it (GL); turano@cerm.unifi.it (PT)

## Abstract

Plasma of COVID-19 patients contains a strong metabolomic/lipoproteomic signature, revealed by the NMR analysis of a cohort of >500 patients sampled during various waves of COVID-19 infection, corresponding to the spread of different variants, and having different vaccination status. This composite signature highlights common traits of the SARS-CoV-2 infection. The most dysregulated molecules display concentration trends that scale with disease severity and might serve as prognostic markers for fatal events. Metabolomics evidence is then used as input data for a sex-specific multi-organ metabolic model. This reconstruction provides a comprehensive view of the impact of COVID-19 on the entire human metabolism. The human (male and female) metabolic network is strongly impacted by the disease to an extent dictated by its severity. A marked metabolic reprogramming at the level of many organs indicates an increase in the generic energetic demand of the organism following infection. Sex-specific modulation of immune response is also suggested.

## Author summary

Metabolites and lipoproteins are the main components of human plasma and their concentration can be determined by nuclear magnetic resonance. In COVID-19 patients there are significant alterations in the concentration of several of these molecules. Using the plasma of more than 600 subjects (510 patients sampled in the acute phase of the infection plus 95 independent recovered subjects), we demonstrate that the dysregulation of these molecules is a function of the disease severity but it is not affected by either the SARS-CoV-2 variants or the vaccination status. The disease signature is particularly evident in those cases that subsequently evolve towards a fatal outcome, and could have prognostic value. Building on this large amount of data, we propose a metabolic reconstruction of the disease using a sex-specific multi-organ metabolic model.

**Data Availability Statement:** This study is available at the NIH Common Fund's National Metabolomics Data Repository (NMDR) website, the Metabolomics Workbench, https://www.metabolomicsworkbench.org where it has been

assigned Study ID ST002404. The data can be
accessed directly via its Project DOI: http://dx.doi.
org/10.21228/M89T2Q.

**Funding:** This work was funded by Regione
Toscana, project COMETA, Bando COVID-19
(https://www.regione.toscana.it/-/bando-ricerca-
covid-19-toscana) to GL and PT. The project also
covers the salaries of VG, WV, VP, TC and NB. The
funders had no role in study design, data collection
and analysis, decision to publish, or preparation of
the manuscript.

**Competing interests:** The authors have declared
that no competing interests exist.

## Introduction

From the year 2020, the resilience of worldwide national health systems was profoundly challenged by the Coronavirus disease 2019 (COVID-19) pandemic caused by the severe acute respiratory syndrome coronavirus 2 (SARS-CoV-2). The viral infection is characterized by a broad spectrum of clinical manifestations from an asymptomatic or pauci-symptomatic disease (in more than 80% of subjects), to interstitial pneumonia and acute respiratory distress syndrome requiring hospitalization and even ventilation of the patients [1–3]. Moreover, it has been demonstrated that a significant subset of patients developed concomitant multi-organ dysfunctions, including acute kidney and liver injuries, thromboembolism and sepsis that contribute to a fatal outcome [4,5]. Additionally, some comorbidities have been proposed as risk factors of disease severity and fatal outcome [6,7]. Although vaccines are now available and have demonstrated high efficacy in decreasing the severity of SARS-CoV-2, vaccination does not prevent SARS-CoV-2 transmission. The diffusion of the different viral variants as well as the phenomenon of the so called "Long-COVID", i.e. long-term effects associated to COVID-19 infection [8], further complicate this scenario.

Massive worldwide efforts by research groups using omics sciences have been made to unravel the disease mechanisms and to identify biomarkers of the disease severity [9–13]. In this framework, [1]H NMR [14–17] plays a role for its ability to reveal a complex blood plasma signature exhibiting the presence of a strong fingerprint of the COVID-19 disease. Highly reproducible alterations of a large number of blood metabolites and lipoprotein parameters were identified as markers of the disease, suggesting the reprogramming of important metabolic pathways aimed at the energy supply for viral replication and for host immunological response [18–29]. However, how the administration of vaccines and the development of the different viral variants impact on the disease fingerprint, remain poorly characterized aspects.

Here, we report a detailed and comprehensive characterization of the metabolomic and lipoproteomic fingerprint of plasma samples of > 500 hospitalized COVID-19 patients, with different disease severities, infected with different viral variants and with different vaccination status. Sex-specific differences as well as the contribution of several comorbidities are also analysed. Our data deeply extend a first metabolomic/lipoproteomic characterization of the disease published at the beginning of 2022 [18], performed on a smaller number of subjects of the same cohort, infected before a significant spread of the δ variant and before widespread COVID-19 vaccination. While confirming previously identified severity markers [18–28], we establish for the first time a correlation between the levels of a few metabolites and lipoproteins and the fatal outcome of the disease. These molecules can therefore be proposed as predictive and prognostic biomarkers. The observed changes are interpreted through simulations of the overall metabolic state of the human body with a recently developed sex-specific multi-organ metabolic model [30], which until now has been used to predict known biomarkers of inherited metabolic diseases in different biofluids. Based on this reconstruction and using as input the changes in the metabolome observed in our cohort, we obtained a comprehensive view of the impact of COVID-19 on the entire human metabolism, which represents a step forward with respect to previous models based on not cell type-specific and not sex-specific flux balance analyses [31].

## Results

In this study, EDTA-plasma samples from 510 COVID-19 positive patients (COVID-19 group) were collected during the period 20/06/2020-17/06/2022. The demographic and clinical data of the cohort are reported in **S1 Table**. This population well represents the incidence of the disease in Tuscan hospitals along the course of the pandemic, in terms of sex, age, severity, and main risk factors/comorbidities. One hundred and fifty-four of these individuals were also

evaluated at a follow-up visit; this group is hereafter named as the "follow-up group", **S2 Table**. The reference group COVID-19-R (**S1 Table**) is constituted by 95 recovered subjects who had contracted the infection during the first wave and did not show any symptom of long COVID at the follow-up visit, when they were sampled. Consistently their metabolite and lipoprotein levels fall within the normality ranges [19].

## Fingerprint of the main COVID-19 variants

To evaluate whether the metabolic and lipoprotein fingerprint of the disease was significantly changed as a function of the different variants of the virus, we selected 3 groups of samples. Based on the data of the statistical incidence of the various SARs-Cov-2 variants in the Tuscany Region (TR) (TR does not systematically determine the variants of each patient through genomic sequencing), only patients infected in periods characterized by defined dominant variants were included. In particular, we considered: i) 87 samples collected when the Wild-Type (WT), α, and β variants were prevalent (01/2020-06/2021), the "wt-α-β group"–these samples are new with respect to those included in our previous study [18]; ii) 91 samples collected when the δ variant was dominant, i.e. the "δ group" (07/2021-12/2021); iii) 44 samples collected when the o variant was dominant (01/2022-02/2022), the "o group". Together, the three groups account for a total of 222 samples. The detailed demographic and clinical characteristics of these subjects are reported in **Fig 1A**.

From the multivariate statistics (see Methods) no clear differences emerge among the metabolomic fingerprints of the three groups of variants; instead, all of them can be almost perfectly discriminated from COVID-19 recovered subjects (**S1 Fig**). The three groups show very similar trends for both metabolite and lipoprotein levels, **Figs 1B–1C and 2**. Conversely, eleven out of 25 metabolites and 16 out of 30 lipoprotein main parameters and main fractions result to be significantly different from the recovered subjects, regardless of the variant, **Figs 1B–1C and 2**.

The three ketone bodies (3-hydroxybutyrate, acetoacetic acid and acetone) along with the amino acids Phe, Met and Ile, the sugars glucose and mannose, and the glycoproteins GlycA and GlycB, are significantly increased in all COVID-19 groups. Citric acid and acetic acid, instead, show decreased levels in all COVID-19 groups. Histidine is the only molecule that shows a different direction of changes among the three groups with respect to the COVID-19-R group; it decreased in the wt-α-β group while it increased in the δ and o groups.

For a few other metabolites (Val, Tyr, lactic acid, creatine, Leu, Gln, formic acid, Ala) we observe the same trends of changes among the three groups, although their variations are not consistently significant. The lack of significance could result from different number of samples in the various groups combined with relatively small fold changes ($|\mathrm{Log_2(FC)}| < 1$).

As for lipoproteins, all the parameters, except free cholesterol (Chol) and apolipoprotein B100 (Apo B100) associated to IDL, show the same trends along all the variants, **Fig 2**. Significantly decreased level of LDL- and HDL-Chol, total Apo A1 and Apo A2, Apo A1-HDL, Apo A2-HDL, triglycerides (TG)-HDL and phospholipids (PhL)-LDL, -IDL and -HDL were observed in all the COVID-19 groups with respect to the recovered subjects. Increased values of the ratio Apo B100/Apo A1 and of TG-LDL are also coherently observed. Moreover, significantly increased levels of free Chol-LDL are observed only in wt-α-β and δ groups, whereas increased levels of the Apo B100-VLDL are observed only in the wt-α-β and o variants. A significant decrement of ApoB-LDL is instead only reported for the δ group. Lipoprotein subfractions and particle numbers are reported in **S2 Fig**.

## Fingerprint of the disease as a function of the vaccination status

Among the 510 COVID-19 positive hospitalized subjects, 71 were vaccinated (with two or more doses of DNA or RNA vaccine, administered at least 15 days before the first positive

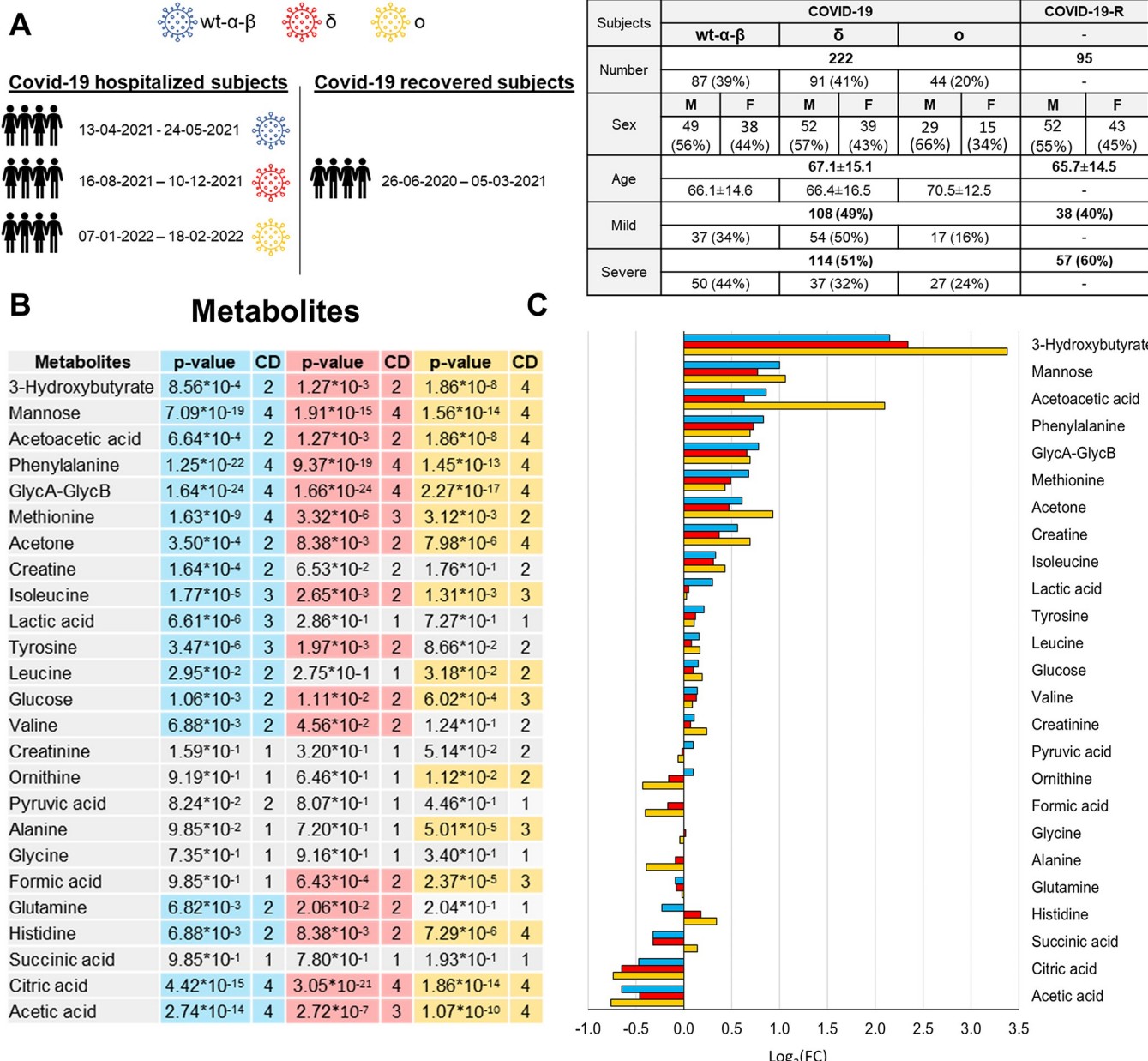

**Fig 1. Metabolomic profiling of the main COVID-19 variants.** (A) Main demographic and clinical characteristics of the enrolled subjects (right panel); for each group of subjects, the sample collection period was reported (left panel). (B) List of the metabolites quantified in plasma samples. The p-values and Cliff's Delta effects size are reported for the comparison between each of the three groups of variants with respect to the COVID-19-R group; p-values <0.05 are highlighted. (C) Values of Log$_2$ fold change (FC) of quantified metabolites. Positive/negative values have higher/lower concentration in plasma samples from each group of variants with respect to the COVID-19-R group. Colour coding: wt-α-β group (cyan); δ group (red); o group (orange).

swab). With the aim of characterizing if the vaccine introduces some important alterations in the metabolomic/lipoproteomic fingerprint of the disease, we compared these samples (hereafter named "VAX group") with the samples of 80 COVID-19 positive patients unvaccinated collected in the same period ("NO-VAX group"). The demographic and clinical data of these groups are reported in **Fig 3A**.

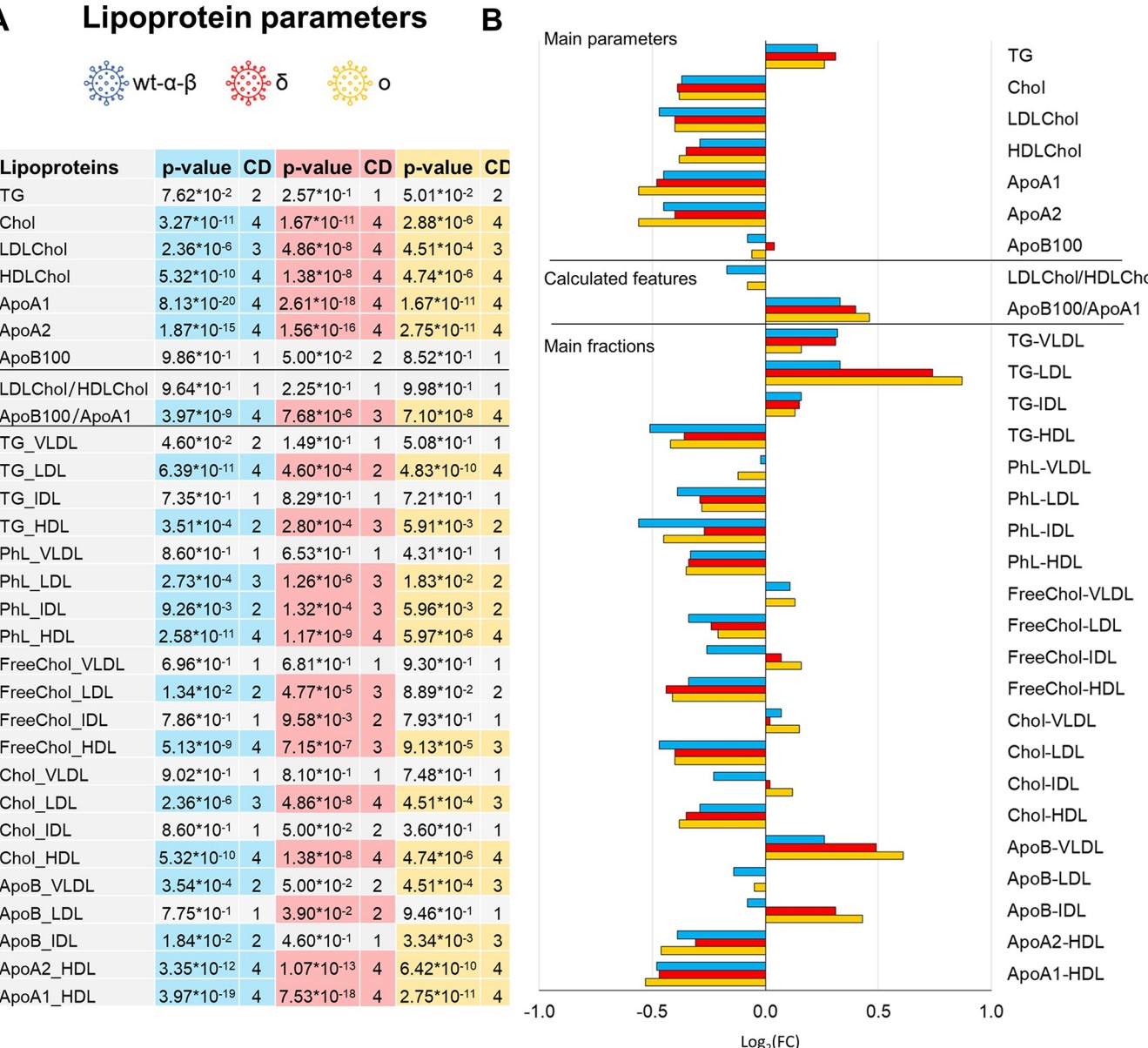

**Fig 2. Lipoproteomic profiling of the main COVID-19 variants.** A) List of lipoprotein parameters (main parameters, calculated features and main fractions) quantified in plasma samples. The p-values and Cliff's Delta effects size are reported for the comparison between each of the three groups of variants with respect to the COVID-19-R group; p-values <0.05 are highlighted. B) Values of Log$_2$ fold change (FC) of quantified lipoprotein parameters (main parameters, calculated features and main fractions). Positive/negative values have higher/lower concentration in plasma samples from each group of variants with respect to COVID-19-R group. Colour coding: wt-α-β group (cyan); δ group (red); o group (orange).

Neither a significant clustering between VAX and NO-VAX groups is observed by multivariate statistics (PCA and RF, **S3 Fig**) nor significantly different levels of metabolites or lipoproteins are found between the two groups. The two groups show the same metabolic and lipoproteomic alterations when compared to the COVID-19-R group (**Fig 3B–3C**). Only succinate shows an opposite trend, being significantly increased when considering the VAX group vs. the recovered subjects and decreased (but not significantly, p-value > 0.05) for the comparison between the NO-VAX group vs. the recovered subjects. Lipoprotein sub-fractions and particle numbers are reported in **S4 Fig**.

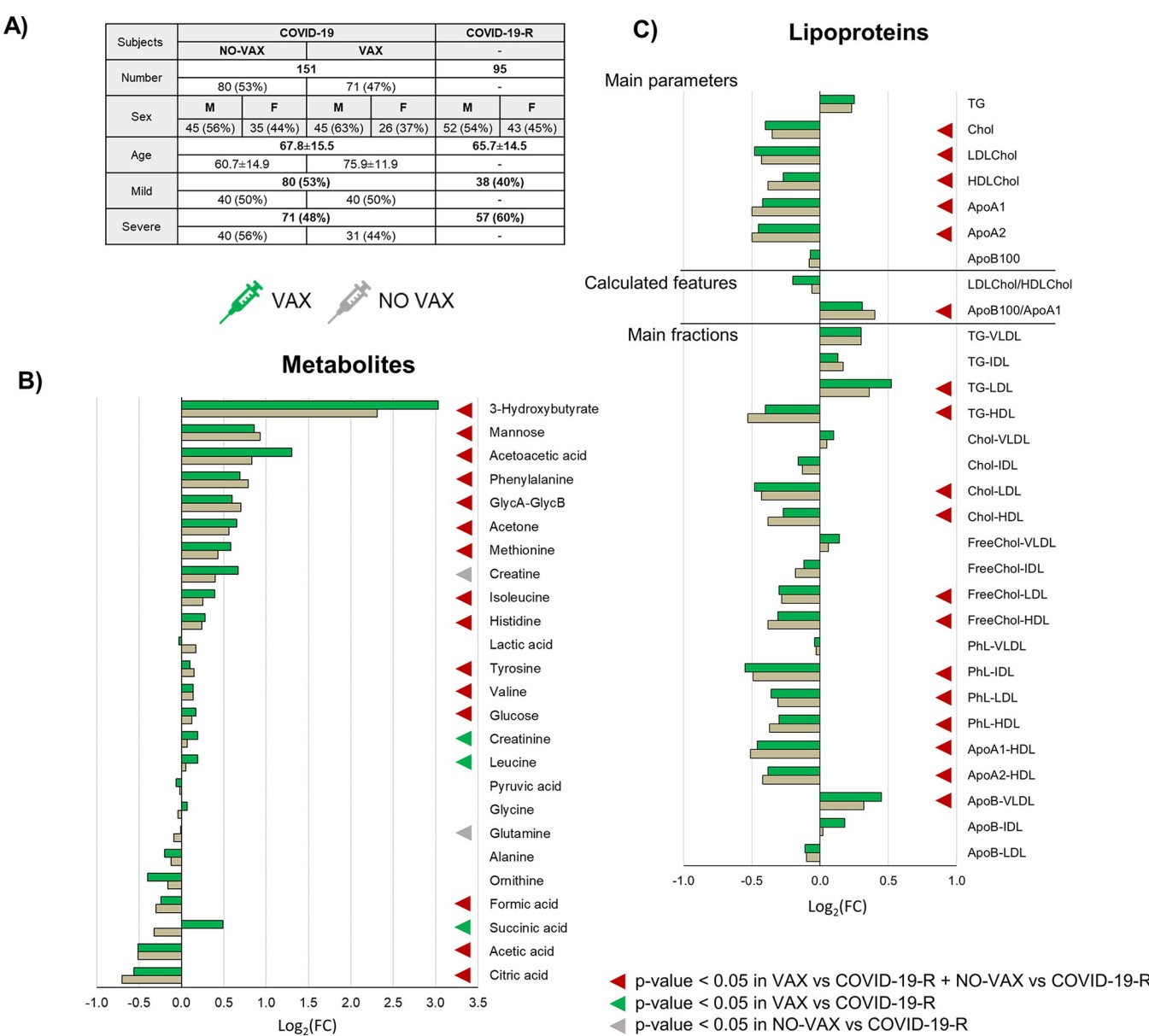

**Fig 3. Metabolomic and lipoproteomic profiling of the VAX and NO-VAX groups.** (A) Main demographic and clinical characteristics of the subjects. (B-C) Values of Log$_2$ fold change (FC) of quantified metabolites (B) and lipoprotein parameters (main parameters, calculated features and main fractions) (C). Positive/negative values have higher/lower concentration in plasma samples from the VAX or NO-VAX groups with respect to COVID-19-R group; p-values <0.05 are highlighted with coloured triangles. Colour coding: VAX group (green); NO-VAX (grey).

## Sex-specific differences in COVID-19 disease

Sex differences in the disease fingerprint are hereafter evaluated. Since our data did not show significant differences in the metabolomic and lipoproteomic fingerprints associated with the main variants of the virus or with the vaccination status, all the 510 samples (274 from the previous study plus 236 newly enrolled) collected from COVID-19 patients are analyzed together, **S1 Table**.

The presence of sex-specific differences in the plasma metabolomic and lipoproteomic profiles of healthy subjects has been characterized in detail [32]. Thus, to extract only COVID-19

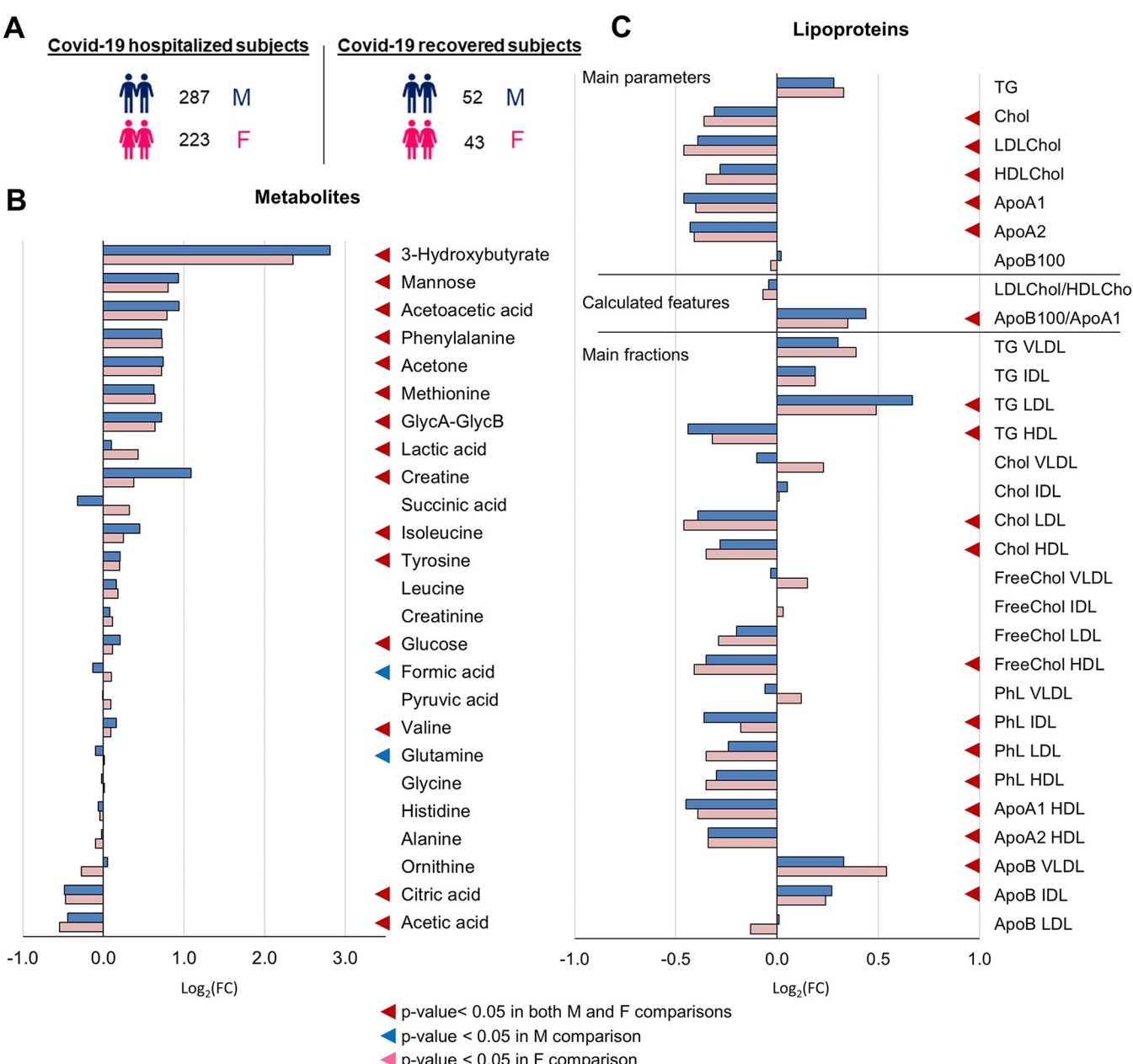

**Fig 4. Sex-related metabolomic and lipoproteomic profiling of COVID-19 subjects.** (A) Number of samples used for the analysis. (B-C) Values of Log$_2$ fold change (FC) of quantified metabolites (B) and lipoprotein parameters (main parameters, calculated features and main fractions) (C). Positive/negative values have higher/lower concentration in plasma samples from the male (M) or female (F) groups with respect to the M or F COVID-19-R group, respectively; p-values <0.05 are highlighted with coloured triangles. Colour coding: male group (blue); female group (pink).

related differences, 287 male and 223 female COVID-19 patients are compared to 52 male and 43 female recovered subjects, respectively, **Fig 4A**.

Again, the strong and solid metabolic fingerprint induced by the infection can be detected in both male and female COVID-19 subjects, **Fig 4B–4C**. No significant differences in the dysregulation trends of either metabolites or lipoprotein parameters emerge between the two groups, with the only exceptions of formic acid and Gln that are slightly but significantly decreased only for male patients. Lipoprotein sub-fractions and particle numbers are reported in **S5 Fig**.

## Global metabolomic/lipoproteomic fingerprint of COVID-19 disease and prognostic biomarkers

Since the previously tested potential confounding factors (variants, vaccination status, sex) did not show any relevant influences on the metabolomic/lipoproteomic fingerprint of the disease, we were prompted to perform a global and comprehensive characterization of the metabolic dysregulation induced by the viral infection. Thus, all the 510 samples collected from the positive subjects are compared to the 95 COVID-19- recovered subjects (**S1 Table**). In parallel, the impact of the disease severity was also evaluated. To this purpose all patients were classified as mild or severe, according to the respiratory symptoms manifested in the acute phase of the infection (see Methods).

Importantly, the RF model built on all samples shows a significant differential clustering between the whole COVID-19 group and the COVID-19-R group with 93.4% accuracy, 94.3% sensitivity, and 88.4% specificity (**Fig 5A**). This comparison highlights 17 significantly dysregulated metabolites (**Fig 5B–5C**) and 77 lipoprotein parameters (**Fig 6**). The levels of 3-hydroxybutyrate, mannose, acetoacetic acid, creatine, Phe, acetone, GlycA-GlycB, Met, Ile, glucose, lactic acid, Val, Tyr, Leu are increased in COVID-19 subjects with respect to the COVID-19-R group; the levels of Gln, citric acid and acetic acid are, instead, decreased. Among them, the concentrations of 8 metabolites result significantly increased/decreased in the comparison between mild and severe COVID-19 subjects (**Fig 5B–5C**). This trend is particularly evident for mannose, phenylalanine, GlycA-GlycB and citric acid (**S6A Fig**). Regarding the lipoprotein panel (**Figs 6 and S6B**), for the main parameters the COVID-19 group is characterized by a significant increment of total TG and a decrement of total Chol, HDL-Chol, LDL-Chol, as well as of ApoA1 and ApoA2. Looking at the level of the main- and sub-fraction composition, the infection leads to: *i*) a decrement of all the components associated with HDL (TG, Ph, Chol, Free Chol, ApoA1 and ApoA2), with HDL3 and, above all, HDL4, as the most affected subfractions; *ii*) an increment of TG-LDL (all subfractions) and a decrement of Chol-, free Chol- and Ph- in all LDL subfractions but LDL2 and LDL6 (LDL4 is the most affected subfraction). *iii*) an increment of TG-VLDL (all subfractions) and of ApoB-VLDL. As in the case of metabolites, several parameters show clear trends according to the disease severity; this is shown in **S6 Fig**, which reports the metabolites and lipoprotein parameters (main parameters, calculated figures and main fractions) that have a p-value < 0.05 and a large Cliff's delta effect size in the comparison between the COVID-19 and COVID-19-R groups, and whose levels are also differentially altered between mild and severe patients.

The presence of clear trends in the concentration levels of some metabolites and lipoproteins according to the disease severity led us to consider the 40 severe subjects (28 wt-α-β, 6 δ, 5 o and 1 not attributable to a specific variant) with a fatal disease as a separate group, hereafter named "fatal group" (**S3 Table**). Interestingly, the levels of glucose, mannose, phenylalanine and of the three ketone bodies (3-hydroxybutyriate, acetone and acetoacetate) significantly increase going from the mild to the fatal group (**Figs 7 and S7**). Mild-to-fatal trends are also observed for the concentration changes of several parameters associated to the lipoprotein sub-fractions HDL4, LDL4, LDL5, LDL6 and VLDL5 (**Figs 7 and S7**). Among all of them, the fatal group shows significantly higher levels of all parameters associated to LDL5 and LDL6 but TG, with LDL5 parameters having the highest Cliff's delta values in the comparison between severe and fatal groups.

## Comorbidities-dependent variations

Finally, the contribution of the patient's comorbidities on the dysregulated levels of metabolites and lipoproteins was also evaluated. **S1 Table** lists the number of subjects affected by the

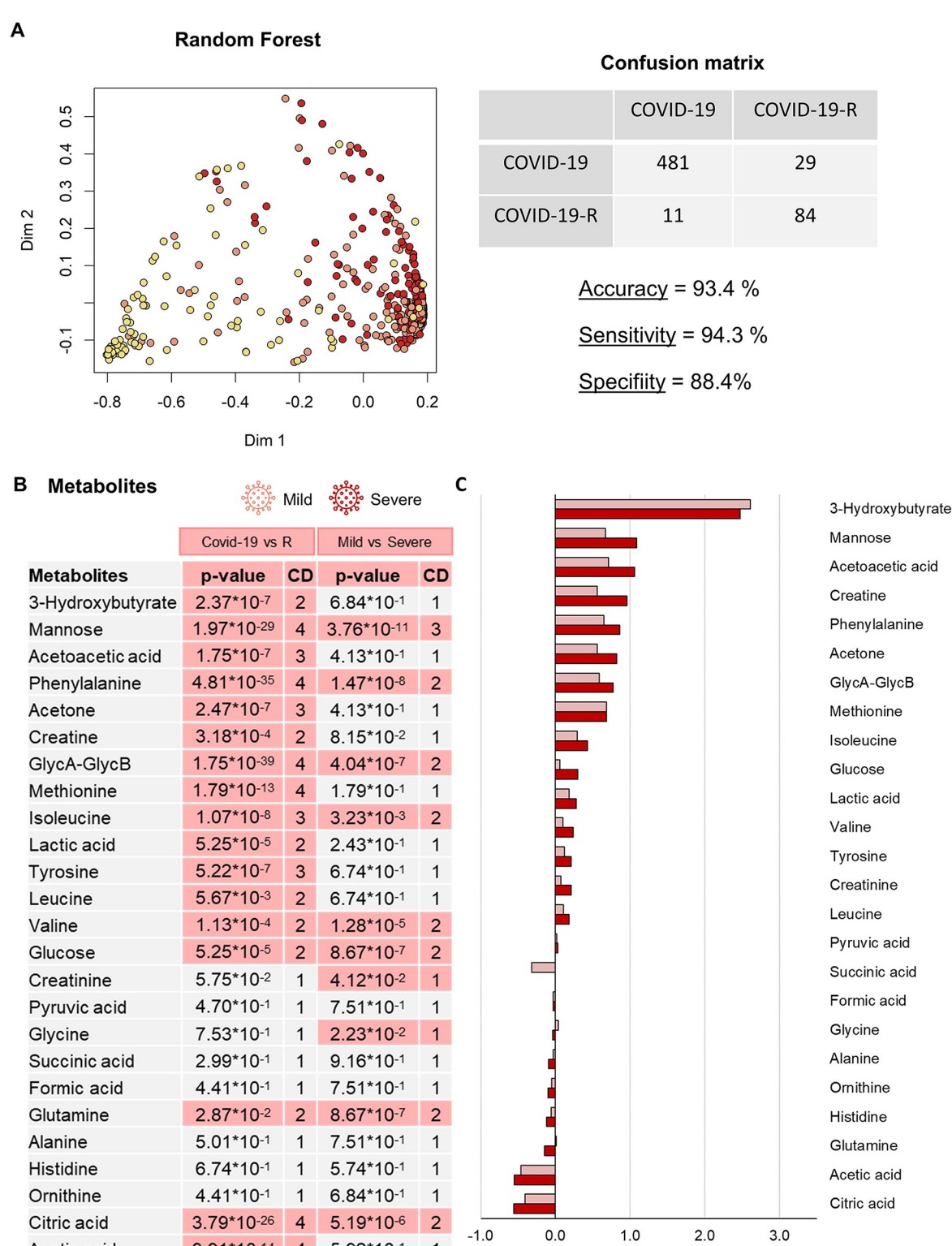

**Fig 5. Metabolomic alterations in COVID-19 patients associated with clinical severity.** (A) Proximity plots of the RF model discriminating COVID-19 patients (mild light red dots, severe red dots), and COVID-19-R subjects (yellow dots) using bucketed NOESY spectra. The confusion matrix with accuracy, specificity and sensitivity values. Notably, the misclassified patients all belong to the mild group. (B) List of the metabolites quantified in plasma samples. The p-values and Cliff's Delta effect-size are reported for the comparison between the COVID-19 group and the COVID-19-R group and for the comparison between mild and severe COVID-19 subjects; p-

values <0.05 are highlighted. (C) Values of Log$_2$ fold change (FC) of quantified metabolites. Positive/negative values have higher/lower concentration in plasma samples from mild or severe subjects with respect to the COVID-19-R group. Colour coding: mild (light red); severe (red).

comorbidities that can be considered as the main risk-factors of severe prognosis [6,7], i.e. asthma or chronic obstructive pulmonary disease (COPD), cardiovascular diseases (such as coronary artery disease (CAD), congestive heart failure (CHF), hypertension, type 2 diabetes (T2DM), dyslipidaemia, chronic kidney disease (CKD) and immune deficiency). Not unexpectedly, subjects affected by T2DM are characterized by higher glucose and mannose concentrations and those affected by CKD have high levels of creatinine (S8 Fig). Whereas creatinine was not found significantly different between the COVID-19 and COVID-19-R groups, high levels of mannose and glucose are very important markers of the COVID-19 signature and even for the discrimination between severe and fatal patients. Importantly, the observed differences in the levels of these two sugars are preserved when diabetic subjects are discarded from the analysis (S9 Fig). It is also important that the trend of increasing concentration for mannose and even more for glucose when going from mild to fatal is more evident for the T2DM subjects that in the no-T2DM patients (S9 Fig).

## The follow-up cohort

For 154 out of 510 COVID-19 patients, a plasma sample was also collected during the follow-up visit, 2–6 months after the first negative swab (follow-up group S2 Table). Thus, for these patients two plasma samples were available, one at the onset of the acute infection (T1) and the second months after the negative swab (T2). They include 111 patients from the wt-α-β group (including 72 subjects with T1 analyzed already in the previous study), 33 from the δ group, 7 from the o group, and 3 not attributable to a specific variant. None of these subjects was diagnosed with long-COVID. The availability of two samples from the same individual allowed us to use a paired test. The same results emerge as from the above-described comparisons between the two independent groups of COVID-19 positive subjects and the COVID-19-R subjects. Consistently, no significant differences in metabolite and lipoprotein concentrations are detected when comparing the T2 samples with those from the COVID-19-R subjects. Indeed, the dysregulated molecules during the infection (i.e., at T1), are essentially reverted to within the normality range in samples collected at T2, S10 Fig. This behaviour is independent of the SARS-CoV-2 variant.

## Genome-scale metabolic modelling

To investigate the impact of COVID-19 on the potential system-level functional metabolic shifts, we contextualized the most recent genome-scale metabolic model (GEM) of the human metabolism with the data obtained in this work. This approach represents a step forward with respect to the available literature data, because it is based on a multi-organ, sex-specific reconstruction. We initially focus on the male model analysis due to the stronger support provided by the higher number of fatal cases with respect to the female ones (27 vs. 13). Following the procedure illustrated in Methods, we generated healthy- (recovered), mild-, severe- and fatal-GEMs and used the underlying (predicted) flux distributions to understand the functional metabolism and metabolic pathways activity that define the severity of the disease. Using the metabolomic data and the most likely contribution of each organ (i.e. inputs and by-products) to the overall blood circulation, we contextualized the human metabolic network.

In analysing the enrichment/depletion of a given metabolite in the blood with respect to a specific organ, a key question arises: is the higher concentration of that specific metabolite in

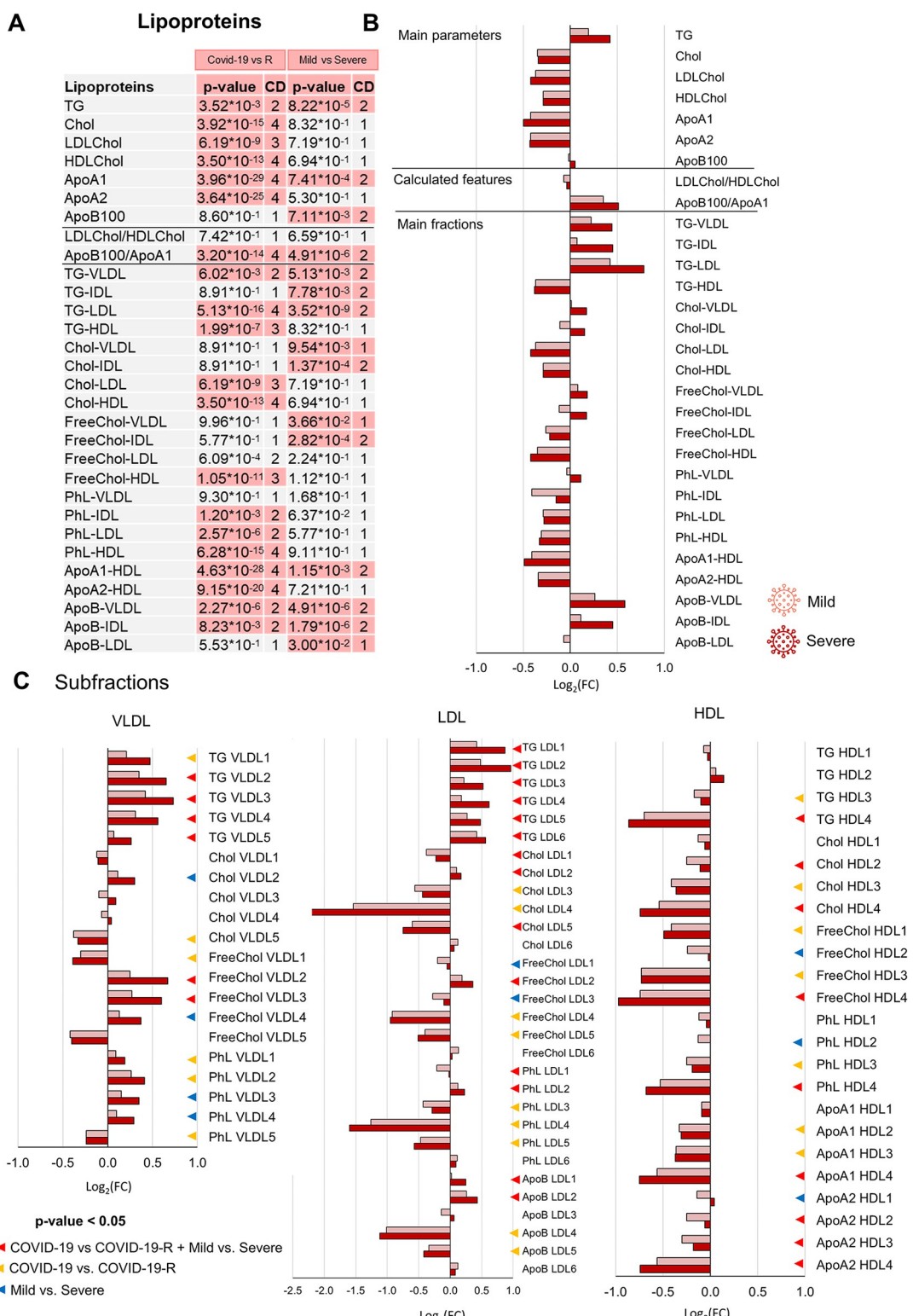

**Fig 6. Lipoproteomic alterations in COVID-19 patients associated with clinical severity.** (A) List of lipoprotein parameters (main parameters, calculated figures and main fractions) quantified in plasma samples. The p-values and Cliff's Delta effect size are reported for the comparison between the COVID-19 group with respect to the COVID-19-R group and for the comparison between mild and severe COVID-19 subjects; p-values <0.05 are highlighted. (B-C) Values of Log$_2$ fold change (FC) of quantified lipoprotein main parameters, calculated features and main fractions (B) and lipoprotein subfractions (C). Positive/

negative values have higher/lower concentration in plasma samples from the COVID-19 group with respect to the COVID-19-R group. Colour coding: mild (light red); severe (red).

the plasma due to the reduced uptake by the corresponding organ or, conversely, it is due to an increased secretion by the same organ? A similar reasoning could be formulated in the case of lower metabolite concentration in the blood that could be the result of an increased uptake or a reduced secretion. We adopted the approach proposed by Dillard et al. (2022)[31] in

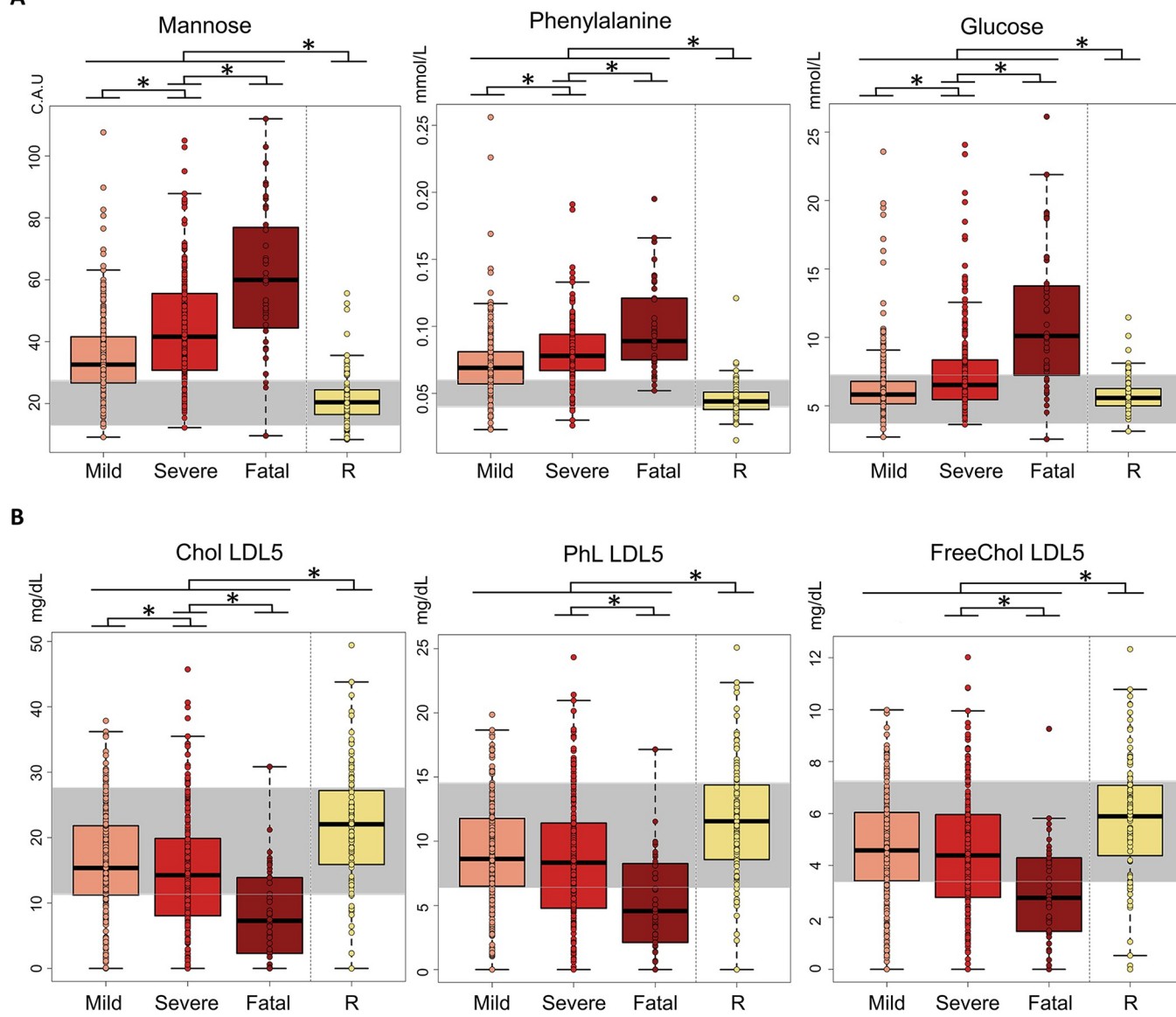

**Fig 7. Markers of fatal events.** Box plots of the concentration levels for (A) metabolites and (B) lipoprotein parameters (main parameters, calculated figures and main fractions) that have a p-value < 0.05 and a large Cliff's delta effect size in the comparison between mild, sever and fatal COVID-19 groups. The concentration levels in COVID-19-R subjects are also reported as control values. In each plot, the grey stripe embraces the concentration range in the reference "healthy" population. Colour coding: mild (light red); severe (red); fatal (dark red) COVID-19-R (yellow). * indicates p-value < 0.05: the upper line indicates the statistical significance between all the COVID-19 subjects and the COVID-19-R group; the lower lines indicate statistical significance between pairs of severity groups.

modelling COVID-19 with not cell-type specific Recon3D and according to which, in an active disease state, increased metabolite production is expected to be more plausible than reduced uptake. Thus, for example, they arbitrarily set to positive exchange bounds (thus simulating secretion) all the model metabolites identified as significantly higher in the severe COVID-19 disease state in order to simulate forced production.

In the first place, we asked whether i) healthy-, mild-, severe- and fatal-GEMs differed among each other and ii) metabolic dysregulation somehow followed the severity of the disease. **Fig 8A** shows that, at least in terms of active/inactive reactions (i.e. reactions carrying/not carrying flux) broad differences exist among the simulated metabolic states. The clustering, however, revealed a pattern that is compatible with a progressive dysregulation that matches the underlying severity of COVID-19 disease (healthy<mild<severe<fatal).

We then focused our attention on those pathways that showed an increase in their activity following the severity of the disease (**Figs 8B and S11**). More in detail, we selected those pathways whose reactions showed, on average, an increase in their flux values from the mild to the fatal states (in **Fig 8B** the healthy state is also included as an additional reference). Among those pathways exhibiting relatively high flux values across all the conditions (lower cluster of

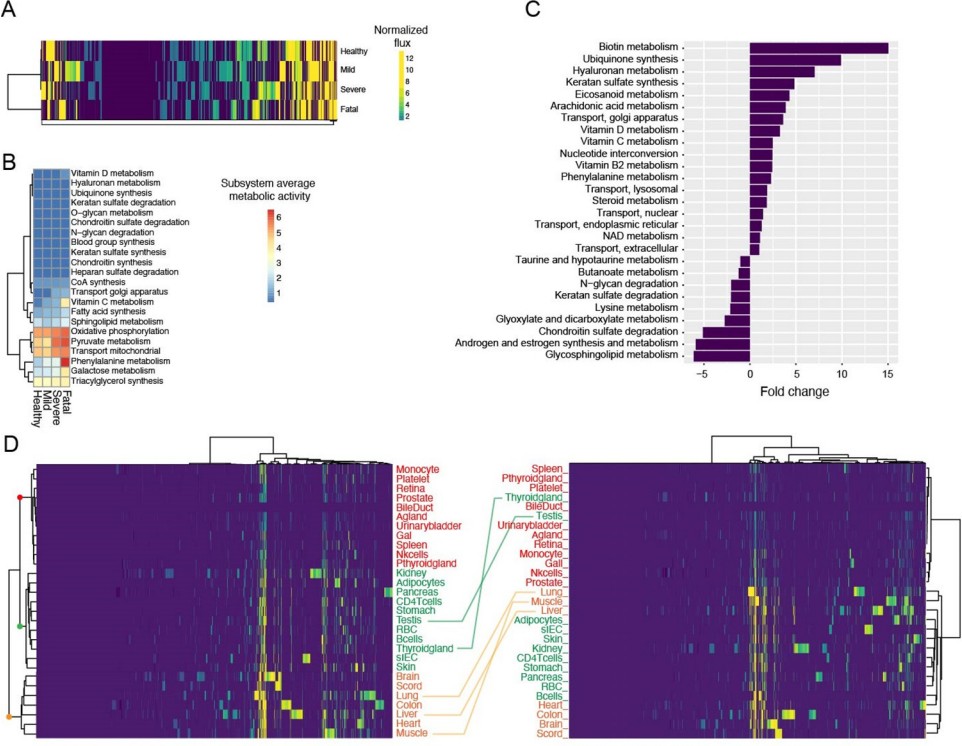

**Fig 8. Genome-scale metabolic modelling of male subjects.** A) Clustering of context-specific models. Each column represents a reaction in the human metabolic network and its color accounts for the (normalized) activity of that specific reaction in the corresponding context-specific model. Models are clustered according to their profile of active/inactive reactions. B) Heatmap that includes those metabolic processes (subsystems) showing an overall flux increase that paralleled that of disease severity (from mild to severe). Their activity in the healthy-model is also included in the figure for clarity. C) Fold change of average metabolic activity for each subsystem in the healthy vs. fatal condition. D) Heatmaps showing the clustering of the different organs according to the activity of each reaction (left healthy state, right fatal state). As in in the panel A, each column represents a reaction in the human metabolic network and its color accounts for the (normalized) activity of that specific reaction in the corresponding organ. In the heatmap on the left, three main clusters are labelled with a red, green and orange dot and the organs belonging to each cluster are labelled accordingly and this color code is maintained in the heatmap on the right. Organs changing clusters in the two heatmaps are connected with a line.

the dendrogram of **Fig 8B**), a few pathways stood out from this analysis, i.e. oxidative phosphorylation and the metabolic pathways of phenylalanine, pyruvate, galactose and triacylglycerol.

The fact that fatal- and healthy-GEMs showed profoundly different metabolic states (**Fig 8A**) and that the activity of some metabolic pathways seems to correlate with COVID-19 severity (**Fig 8B**), prompted us to compare these two flux distributions and find those pathways (possessing at least five metabolic reactions) that differ in the two most extreme conditions (healthy vs. fatal **Fig 8C**). This analysis revealed the presence of 18 and 9 pathways that increase and decrease their activity in the fatal state with respect to the healthy one, respectively. The top three metabolisms that showed an increase in the fatal conditions with respect to the healthy one included biotin metabolism, ubiquinone synthesis and hyaluronan metabolism. On the contrary, the pathways that showed a marked decrease in their activity with respect to the healthy condition were glycosphingolipid metabolism, androgen and estrogen metabolism and chondroitin sulphate degradation.

The organ-specific structure of the human metabolic model adopted here offered the possibility to investigate the impact of COVID-19 on the metabolism of each organ represented therein. We thus clustered the different organs based on the activity of their metabolic reactions in the healthy and fatal states (**Fig 8D**). This analysis revealed a clear cut between three main clusters whose structure is overall conserved in the two extreme states. The first (red cluster in **Fig 8D**) embeds somehow peripheral organs, while the other two clusters (labelled with green and orange colours) include, among the others, all the vital organs (that are mostly included in the orange cluster, **Fig 8D**). Importantly, considering the healthy state (left heatmap of **Fig 8D**), the fluxes in the first clusters appear, on average, lower than in the second one, suggesting a lower metabolic activity in these organs with respect to the ones included in the second cluster. In the fatal metabolic state (right heatmap of **Fig 8D**) this situation appears to be more pronounced, with key organs showing an overall higher activity with respect to those embedded in the red cluster, compatibly with the disease state that characterized these patients. More in general, besides the conservation of the main clustering topology that subdivides the organs in three main clusters, we noticed a marked reshuffling of the relative order of each organ-specific metabolic network within these clusters, especially for some organs (thyroid gland, testis, lung, liver and muscle). From an analogous comparison between severe and fatal states (**S11 Fig**), we can establish that the main contribution to the metabolic dysregulation is associated to the progression from severe illness to the fatal state.

Also, we compared the sex-specific models obtained by integrating male and female models on the corresponding metabolic reconstructions. The overall structure of the models was maintained between sexes, both when considering the overall flux distribution (**Figs 8A and S12**) and the organ-level metabolic activity (**Figs 8D and S12**). Pathways increasing their activity concordantly with disease severity in both male and female models included N-glycan degradation, keratan sulphate synthesis, oxidative phosphorylation, and phenylalanine metabolism. We also found a pool of subsystems that showed a positive fold-change (i.e. higher metabolic activity) in the comparison between healthy and fatal states, both in male and female models. These were ubiquinone synthesis, vitamin D, C and B2 metabolism, and eicosanoid metabolism. Interestingly, we also found pathways that displayed sex-specific increase in their activity (e.g. arachidonic acid metabolism for male models and fatty acid oxidation, squalene and cholesterol synthesis in female models) or that showed opposite trends in the sex-specific simulations (e.g. androgen and estrogen synthesis and metabolism, decreasing in males and increasing in females).

## Discussion

In this study we used an untargeted $^1$H NMR approach to assess the metabolomic/lipoproteomic changes in the EDTA-plasma samples of a large Italian cohort of subjects (the COMETA cohort) with mild to fatal COVID-19. The advantages of using NMR with respect to the more sensitive mass spectrometry approach consist in being untargeted, highly reproducible, intrinsically quantitative and, in the case of blood derivative analyses, in allowing the automated profiling of a large ensemble of lipoprotein subfractions.

In a previous study from this consortium, 274 samples of the COMETA cohort were collected from COVID-19 patients hospitalized during the first wave of the pandemic (06/2020-04/2021), i.e. before a significant spread of the δ variant and before a massive vaccination campaign [18]. Here, samples from a much larger cohort (236 additional patients in the acute phase of the infection) and collected in a more expanded time frame (06/2020-06/2022) became available; they include 87 new patients from the wt-α-β group; 91 from the δ group; 44 from the o group, and 14 not attributable to a specific variant. Together, they form a cohort of 510 patients. The new analyses contribute to the advancement of the state of the art in three main aspects:

1. We demonstrate that the metabolomic/lipoproteomic signature of the disease is robust and does not change in a large and heterogenous cohort of patients, sampled during the different waves of COVID-19 infection corresponding to the spread of different variants and having different vaccination status. Interestingly, most metabolites and lipoprotein parameters show the same trends of changes along all the variants. The larger increase in ketone bodies observed for the o variant can be attributed to the higher percentage of fatal events in this subgroup. Histidine, together with IDL-associated Chol and Apo B100 are the only molecules that increase in the δ and o groups but decrease in the wt-α-β group. No significant correlations (Pearson test) were found among these molecules. The decrement of histidine was already reported also in the other metabolomic studies based on samples collected before the spread of the δ variant [19,20,27]; to our knowledge no detailed information is yet available for the other variants. The signature of the disease is completely independent from the vaccination status of the subjects; indeed, infected vaccinated subjects show the same metabolic and lipoproteomic changes observed for non-vaccinated subjects (with the only notable exception of succinate, which need to be further analyzed in future studies). Independence of the metabolomic profile from the vaccination status has recently been mentioned by other authors [29]. Notably, all the VAX subjects here considered had been vaccinated since more than 2 weeks, thus ensuring that the specific changes of the lipoproteome observed following vaccination had become negligible [33].
The metabolomic/lipidomic signature of the disease does not show any relevant difference between males and females, indicating that the strong disease signature overcomes any sex-specific signature. The only relevant variation between the two sexes concerns a decrease in formate and Gln in male patients; these two molecules are known to be important for the modulation of the immune response [34,35].
Lastly, we could also confirm in a set of 154 subjects sampled at 2 different time points that, after 2–6 months from negativization, in the absence of any clinical long-term symptoms, the dysregulated features all revert toward their normal value, indicating the substantial metabolic healing of COVID-19 clinically-recovered subjects.

2. The metabolomic and lipoproteomic changes induced by the SARS-CoV-2 infection that are described here are largely coherent with those obtained in our previous studies and also observed by other research groups worldwide [18–29]. The high reproducibility among all

the published data demonstrates that they accurately reflect pathophysiological changes associated to COVID-19. Additionally, here we can discriminate subjects who will progress towards a fatal outcome based on the levels of some molecules, that can be considered as prognostic markers. These subjects are characterized by the highest plasma levels of mannose and phenylalanine but also by very high levels of glucose and of the three ketone bodies, along with the largest dysregulation of several lipoprotein parameters. Notably, the highest fold changes for both glucose and mannose are observed for patients with T2DM, a pre-existing comorbidity identified as an important risk factor for COVID-19 associated mortality [7,36].

3. In the literature, the overall metabolic changes observed in COVID-19 have been tentatively explained in terms of a complex host-virus interaction aimed at the energy supply for both viral replication and host immunological response, leading to a dysregulation of carbohydrate and lipid metabolism and to mitochondrial dysfunction [18–28]. Also, metabolic modelling has shown to be a useful tool to systemically explore the metabolic impact of COVID-19 and to prioritize specific metabolic reactions in the search of novel treatment strategies [31,37–41]. The release in 2021 of the first multi-organ and sex-specific genome-scale human metabolic reconstruction offered us the possibility to i) more accurately constraint the model in a knowledge-based way; explore ii) the metabolic impact of COVID-19 on the entire metabolism and iii) the sex-dependent metabolic response to the disease. Based on NMR metabolomic data, our reconstruction shows that, globally, the human (male and female) metabolic network is strongly impacted by the disease, and that the extent of changes in metabolic configuration follows the disease severity (**Fig 8A**). Also, we found a marked metabolic reprogramming at the level of many organs embedded in our reconstruction. Indeed, despite the three main clusters that were conserved in the two cases (red, green, and orange clusters in **Fig 8D**) contained the same organs, i) the relative order of this clustering is different in the two conditions (red and green clusters clustering together in the healthy condition but not in the fatal one) and ii) the relative position of the organs inside the same cluster differ significantly in the comparison between healthy and fatal heatmap. Functionally speaking, a few pathways occurred multiple times, both among those showing an overall activity trend that mirrored disease severity, and among those that showed the highest fold change in the healthy vs. fatal comparison. These included oxidative phosphorylation and the metabolic pathways of phenylalanine, pyruvate, galactose, and triacylglycerol. Remarkably, i) these pathways include the possible disease markers experimentally identified by our NMR approach (e.g. phenylalanine and mannose) and ii) a large body of literature identified perturbations of the metabolites belonging to these pathways as possible metabolic responses to the COVID-19 disease state [42–46]. Our data also suggest the presence of system-level differences in the metabolic network of males and females in response to COVID-19 disease. A few pathways, whose changes were consistent between sexes (ubiquinone synthesis, vitamin D, C and B2 metabolism and eicosanoid metabolism), indicate an increase in the generic energetic demand of the organism following the severity of the disease (for both viral replication and host immunological response). Many other pathways showed marked sex-related differences. This is the case for example of the metabolic activities of reproductive steroid hormones that, in our sex-specific simulations, showed opposite trends (decreasing in males and increasing in females, **Figs 8C and S12**, respectively). Importantly, the immunomodulatory effects of estrogens in different viral infections are largely known [47], as well as their involvement in the response to COVID-19 [48,49]. Similarly, phenylalanine metabolism (despite following the same trend of disease severity in both males and females) was shown to be more active in men with respect to

women in healthy vs. fatal contrast (**Figs 8C and S12C**). This confirms previous findings on the sex dependency of this metabolic pathway during COVID-19 disease and on its possible connection with increased inflammatory status [50,51]. On the one hand, these findings highlight the importance of these specific metabolic pathways for monitoring disease progression, and, on the other hand, they also promote the use of systems biology tools for identifying non-trivial consequences of metabolic changes at the whole-body level.

## Materials and methods

### Ethics statement

The study was conducted in accordance with the Declaration of Helsinki; it was approved by Comitato Etico Regionale per la Sperimentazione Clinica della Toscana—sezione Area Vasta Centro, code "18436_bio". Written informed consent for inclusion was obtained from each subject before enrolment in the study.

### Patients' recruitment and sample collection

All the subjects included in this study were enrolled in the framework of the COMETA project, funded by the Tuscany Region, Italy. A total of 605 subjects (**S1 Table**) were recruited at the Santa Maria Nuova hospital of the Azienda USL Toscana Centro, in Florence (Italy) in the period between 20/06/2020 and 17/06/2022, i.e. during the first three waves of the COVID-19 pandemic in Italy. They include two different groups of subjects, namely:

i. 510 COVID-19 hospitalized patients, that resulted positive for SARS-CoV-2 infection (with molecular nasopharyngeal swab), infected with different variants of the virus and with various disease severity; for all them a blood sample was collected during the acute phase of the infection. Additionally, for 154 out of 510 COVID-19 patients, we collected a second plasma sample during a follow-up visit (2–6 months after the first negative swab). This group is named as the "follow-up group", **S2 Table**.

ii. 95 COVID-19 recovered subjects (COVID-19-R group), previously hospitalized for SARS-CoV-2 infection (all during the first wave of pandemic); the blood of these subjects, who do not show any symptoms of persistent illness and could be considered as fully recovered, was collected 2–6 months after test negativization, during the follow-up visit (for this group we do not have the samples at the moment of the infection).

COVID-19 patients were classified as mild or severe according to the respiratory symptoms in the acute phase of the infection. All the subjects not requiring treatment with oxygen (or not requiring supplemental oxygen with respect to the treatment in progress before infection) or requiring oxygen treatment mask (Ventimask, VM) or nasal prongs with $FiO_2 \leq 40\%$ were classified as mild; patients requiring non-invasive ventilation (NIV) or MV with high $FiO_2 > 40\%$ or requiring orotracheal intubation (OTI) were classified as severe. Blood withdrawal occurred before NIV / VM ($FiO_2 > 40\%$) or OTI. Forty of these patients had a fatal outcome (**S3 Table**).

For COVID-19 recovered subjects, blood samples have been collected at the moment of admission to the hospital, when patients were swabbed regardless of the cause of hospitalization, and before the start of any therapy or other medical interventions (oxygen mask, ventilation, parenteral nutrition).

All plasma samples were collected, processed and stored according to ISO standards (ISO 23118: 2021), designed for high quality biological samples for metabolomic analysis [52–54].

The metabolomic analyses were performed on EDTA-plasma samples. Before analysis, the EDTA-plasma samples were stored at -80°C in the repository of the da Vinci European Biobank, which offered a conservation service (daVEB, DOI: 10.5334/ojb.af, https://www.unifi.It/vp-11370-da-vinci-european-biobank.html, Italy).

## NMR analysis and spectral processing

NMR samples were prepared and recorded according to standard procedures for serum/plasma samples for metabolomics analysis [14,15].

NMR spectra for all the samples were acquired using a Bruker 600 MHz spectrometer (Bruker BioSpin) operating at 600.13 MHz of Larmor proton frequency and equipped with a PATXI $^1$H – $^{13}$C– $^{15}$N and $^2$H decoupling probe including a z-axis gradient coil, automatic tuning-matching (ATM) and an automatic, refrigerated sample changer (SampleJet, Bruker BioSpin). A BTO 2000 thermocouple served to stabilize the temperature to a level of approximately 0.1 K in the sample. Before measurement, the samples were kept for 5 minutes inside the NMR probe head, for temperature equilibration at 310 K. For each sample, three one-dimensional $^1$H NMR spectra were acquired with water peak suppression and different pulse sequences [14]: i) standard NOESY 1Dpresat ii) standard 1D CPMG iii) standard 1D diffusion-edited. The parameters of each experiment are reported in **S4 Table**. Free induction decays were multiplied by an exponential function equivalent to a 0.3 Hz line-broadening factor before applying Fourier transform. Transformed spectra were automatically corrected for phase and baseline distortions and calibrated at the glucose doublet at δ 5.24 ppm using TopSpin 4.1 (Bruker BioSpin).

## Statistical analysis

All the statistical analyses were performed using the R software (R. 3.0.2).

The multivariate analyses were applied on NOESY binned spectra. To this aim, each spectrum was segmented into 0.02 ppm chemical shift bins, from 10.00 to 0.2 ppm, with the exclusion of EDTA resonances (regions: 2.53–2.60, 2.68–2.73, 3.07–3.24, 3.58–3.64 ppm) and water signal (region: 4.40–5.00 ppm); the corresponding spectral areas were integrated (Assure-NMR software, Bruker BioSpin). Unsupervised Principal Component Analysis (PCA) was used as an exploratory analysis to obtain a preliminary outlook of the data (visualization in a reduced space, presence of clusters or outliers).

Different types of multivariate statistics (i.e. Random Forest (RF) and Orthogonal Projections to Latent Structures-Discriminant Analysis (OPLS-DA)), were tested for supervised classification. Despite, the obtained models were essentially independent of the multivariate method used, here we reported the results obtained by RF algorithm because it allows stratified samplings, to ensure equal representation in the case of unbalanced groups [14].

The accuracy, sensitivity, and specificity of all calculated models were assessed according to the standard definitions.

Twenty-five metabolites, were assigned in all the spectra and their concentrations analysed. The assignment procedure was performed using an $^1$H NMR spectra library of pure organic compounds (BBIOREFCODE, Bruker BioSpin), public databases, e.g. the Human Metabolome Database (https://hmdb.ca/), and stored reference $^1$H NMR spectra of metabolites. Metabolites were analysed using the In Vitro Diagnostics research (IVDr) B.I.-Quant PS tool (Bruker, BioSpin). For metabolites that are not present in the IVDr list, the respective areas were integrated using a R script developed in-house. The IVDr Lipoprotein Subclass Analysis B.I.-LISA tool (Bruker, BioSpin) was used to extract one hundred fourteen parameters associated to lipoproteins (main parameters, calculated features, main fractions, sub-fractions and particle numbers)

[55]. As done in a previous work from the COMETA consortium [18], a reference population of EDTA-plasma samples from 177 (86 males and 91 females) healthy subjects was used to calculate the "healthy" deviation range (mean ± SD) for each metabolite and lipoprotein.

The non-parametric Wilcoxon-Mann-Whitney was used to determine the significantly different parameters between the different groups of subjects. The obtained p-values were adjusted for multiple tests using False Discovery Rate Correction (FDR) according to the Benjamini-Hochberg method; an adjusted p-value <0.05 was considered statistically significant. Effect size (Ef) was also calculated; the magnitude is assessed using the thresholds provided in Romano et al. [56], that is |Ef| <0.147 "negligible—1", |Ef| <0.33 "small—2", |Ef| <0.474 "medium—3", otherwise "large—4".

## Genome-scale metabolic modelling

Metabolomics data were integrated with the most recent human GEM reconstruction, which is based on the integrated metabolism of 26 main organs of the human body and 6 blood cells types and provides sex-specific whole-body metabolism; in practice, two reconstructions are currently available "Harvey" and "Harvetta" accounting for the male and female metabolic models, respectively [30]. Here we apply this reconstruction to interpret the metabolic changes observed in plasma as a function of COVID-19 severity. We took advantage of the fact that in the updated human reconstruction the values of the exchange reactions (i.e. those reactions that account for the uptake/release of the nutrients by each organ) are the result of extensive literature search and accurately define the metabolic exchanges of each organ with the overall blood circulation. Thus, the direction of the exchange reactions (uptake or release) were set according to Thiele et al. (2020) [30]. The extent of the exchanges (i.e. the rates) were instead defined according to the metabolomic data obtained in this work. More in detail, we post-processed the metabolomic dataset as follows. The metabolomic data were first divided between male and female patients in order to be able to integrate each of the two datasets in the corresponding sex-specific metabolic reconstruction and then divided on the basis of the severity condition (recovered, mild, severe and fatal); as for the rest of this study and in accordance to previous findings [18], the recovered subjects are taken as healthy controls. Then, for each metabolite/lipoprotein, the mean among each sample and the fold changes (FC) among each of the following contrasts was computed: recovered vs. mild, recovered vs. severe, recovered vs. fatal. This latter step was performed using the *foldchange* function of the R package *gtool*. The Wilcoxon test was then used to assess the significance of the changes among the different conditions and the R function *p.adjust* was used to compute the False Discovery Rate (FDR) values. The FC of the plasma metabolites/lipoproteins that resulted to be statistically different in the examined conditions were used to constraint the model as described below.

Using COBRApy [57] (version 0.26) and selecting the Gurobi solver (version 9), we used Flux Balance Analysis (FBA) to create a standard "healthy" flux distribution by leaving the model unconstrained, simulating a Mediterranean diet and optimizing for the biomass objective function of the model. The other flux distributions (mild, sever and fatal) were computed modifying the upper and lower boundaries (UB and LB, respectively) using plasma metabolites FCs. More in details, for each $i_{th}$ exchange metabolites, we set the UB and LB as follows:

$$UB > 0 \land LB < 0 \begin{cases} FC > 0; & x_{ub} = BV \cdot FC \\ & x_{lb} = BV \cdot 1/FC \\ FC < 0; & x_{ub} = BV \cdot 1/|FC| \\ & x_{lb} = BV \cdot |FC| \end{cases}$$

$$UB \wedge LB < 0 \begin{cases} FC > 0; \ x_{ub} \wedge x_{lb} = BV \cdot 1/FC \\ FC < 0; \ x_{ub} \wedge x_{lb} = BV \cdot |FC| \end{cases}$$

$$UB \wedge LB > 0 \begin{cases} FC > 0; \ x_{ub} \wedge x_{lb} = BV \cdot FC \\ FC < 0; \ x_{ub} \wedge x_{lb} = BV \cdot 1/|FC| \end{cases}$$

Where $UB_i$ and $LB_i$ represents the upper and lower boundaries for the $i_{th}$ metabolite of all the exchange metabolites in the reconstruction, FC represents the fold change of the $i_{th}$ metabolite in a specific contrast, $x_{ub}$ and $x_{lb}$ represent the actual upper and lower values of the constrained reaction in the model after metabolomic data integration, $BV$ represents the default boundary value in the original model [30], respectively. From the four hypothetical flux distributions that we obtained (healthy, mild, severe and fatal) we pruned those reactions that display no flux in any of the four conditions. We used the remaining reactions to investigate the metabolic changes at the whole-body level that better account for the measured metabolic changes. To efficiently summarize our data, we took advantage of the model-embedded association between each reaction and its corresponding subsystem (i.e. cellular process) and computed a subsystem average metabolic activity (S.A.M.A.) as follows:

$$S.A.M.A. = \frac{\sum_{i=1}^{N} |f_i|}{N}$$

Where $f_i$ represents the function of the $i_{th}$ of the $N$ reactions included in a given subsystem.

Similarly to what described above for each metabolite in the metabolomic dataset, FC and p-values were computed for each subsystem in the contrasts recovered-mild, recovered-severe and recovered-fatal. Heatmaps were reconstructed using *pheatmap*, and selecting the Ward's (*ward.D*) clustering method.

## Supporting information

**S1 Table. Demographic and clinical characteristics of the COMETA cohort.** List of abbreviations: M: Males; F: Females; COPD: Chronic Obstructive Pulmonary Disease; CAD: Coronary Artery Disease; CHF: Congestive Heart Failure; T2DM: Type 2 Diabetes; CKD: Chronic Kidney Disease.
(XLSX)

**S2 Table. Demographic and clinical characteristics of the follow-up group.** List of abbreviations: M: Males; F: Females; COPD: Chronic Obstructive Pulmonary Disease; CAD: Coronary Artery Disease; CHF: Congestive Heart Failure; T2DM: Type 2 Diabetes; CKD: Chronic Kidney Disease.
(XLSX)

**S3 Table. Demographic and clinical characteristics of the COVID-19 patients with a fatal outcome.** List of abbreviations: M: Males; F: Females; COPD: Chronic Obstructive Pulmonary Disease; CAD: Coronary Artery Disease; CHF: Congestive Heart Failure; T2DM: Type 2 Diabetes; CKD: Chronic Kidney Disease.
(XLSX)

**S4 Table. Parameters of the NMR experiments acquired for each plasma-EDTA samples at 600 MHz spectrometers.**
(XLSX)

**S1 Fig. Multivariate analysis of the main COVID-19 variants.** A) PCA Score Plot based on bucketed NOESY spectra of the main three COVID-19 variant groups and COVID-19-R group. B) Proximity plots of the RF model discriminating the COVID-19 variant groups and COVID-19-R subjects using bucketed NOESY spectra. The confusion matrix and the accuracy value are reported. Colour coding: wt-α-β group (cyan); δ group (red); o group (orange); COVID-19-R (yellow).
(TIF)

**S2 Fig. Lipoproteomic alterations (subfractions and particle numbers) in the main COVID-19 variants.** Values of $Log_2$ fold change (FC) of quantified lipoprotein parameters (particle numbers and HDL, LDL, and VLDL subfractions). Positive/negative values have higher/lower concentration in plasma samples from each of the three variant COVID-19 groups with respect to COVID-19-R group. p-values <0.05 are highlighted with colored squares. Colour coding: wt-α-β group (cyan); δ group (red); o group (yellow).
(TIF)

**S3 Fig. Multivariate analysis of the VAX and NO-VAX groups.** A) PCA Score Plot based on bucketed NOESY spectra. B) Proximity plots of the RF model discriminating the COVID-19 VAX group and the COVID-19 NO-VAX group using bucketed NOESY spectra. The confusion matrix and the accuracy value are reported. Colour coding: VAX group (green); NO-VAX group (grey).
(TIF)

**S4 Fig. Lipoproteomic alterations (subfractions and particle numbers) in the VAX and NO-VAX groups.** Values of $Log_2$ fold change (FC) of quantified lipoprotein parameters (particle numbers and HDL, LDL, and VLDL subfractions). Positive/negative values have higher/lower concentration in plasma samples from the VAX or NO-VAX groups with respect to the COVID-19-R group; p-values <0.05 are highlighted with coloured triangles. Colour coding: VAX group (green); NO-VAX (grey).
(TIF)

**S5 Fig. Sex-related lipoproteomic alterations (subfractions and particle numbers) of COVID-19 subjects.** Values of $Log_2$ fold change (FC) of quantified lipoprotein parameters (particle numbers and HDL, LDL, and VLDL subfractions). Positive/negative values have higher/lower concentration in plasma samples from male (M) or female (F) groups with respect to M or F COVID-19-R subjects, respectively; p-values <0.05 are highlighted with coloured triangles. Colour coding: male group (blue); female group (pink).
(TIF)

**S6 Fig. Markers of clinical severity.** Box plots of the concentration levels for (A) metabolites and (B) lipoprotein parameters (main parameters, calculated figures and main fractions) that have a p-value < 0.05 and a large Cliff's delta effect size in the comparison between COVID-19 and COVID-19-R groups and whose levels are also significantly altered between mild and severe patients. In each plot, the grey stripe embraces the concentration range in the reference "healthy" population. Colour coding: mild (light red); severe (red); COVID-19-R (yellow). *indicates p-value < 0.05: the upper line indicates the statistical significance between all the COVID-19 subjects and the COVID-19-R group; the lower line indicates statistical

significance between mild and severe subjects.
(TIF)

**S7 Fig. Markers of fatal events.** Box plots of (A) metabolites and (B) lipoproteins concentration levels in COVID-19 positive subjects grouped according to disease severity. The concentration levels in COVID-19-R subjects are also reported. Colour coding: mild (light red); severe (red); fatal (dark red); recovered subjects (yellow). * indicates p-value < 0.05: the upper line indicates the statistical significance between all the COVID-19 subjects and the COVID-19-R group; the lower lines indicate statistical significance between pairs of severity groups.
(TIF)

**S8 Fig. Comorbidities-dependent variations.** Box plots of Mannose, Glucose and Creatinine concentration levels in the 510 COVID-19 positive subjects grouped as a function of the main comorbidities. The concentration levels in COVID-19-R subjects are also reported as control values (yellow bar). In each plot, the grey stripe covers the concentration range in a "healthy" population. * indicates p-value < 0.05: the upper line indicates the statistical significance between all the COVID-19 subjects and the COVID-19-R group; the lower line indicates that at least one of the comorbidity groups (circled in red) is significantly different from all the others. List of abbreviations: COPD: Chronic Obstructive Pulmonary Disease; CAD: Coronary Artery Disease; CHF: Congestive Heart Failure; T2DM: Type 2 Diabetes; CKD: Chronic Kidney Disease.
(TIF)

**S9 Fig. Comorbidities-dependent variations of severity markers.** Box plots of (A) mannose and (B) glucose concentration levels in COVID-19 positive subjects grouped according to the grade of the disease severity. The concentration levels in COVID-19-R subjects are also reported. Left panels: all the subjects; middle panels: the T2DM subjects were excluded from the analysis; right panels: only T2DM subjects. In each plot the grey stripe covers the concentration range in a "healthy" population. Colour coding: mild (light red); severe (red); fatal (dark red); recovered subjects (yellow). * indicates p-value < 0.05: the upper line indicates the statistical significance between all the COVID-19 subjects and the COVID-19-R group; the lower lines indicate statistical significance between pairs of severity groups.
(TIF)

**S10 Fig. Time-dependence of the severity markers in the follow-up group (S2 Table).** Box plots of the concentration levels for (A) metabolites and (B) lipoproteins that have a p-value < 0.05 and a large Cliff's delta effect size in the comparison between the plasma samples collected at the moment of the acute infection (T1) and the samples collected at the follow-up visit (T2). The concentration levels in COVID-19-R subjects are also reported as control values. In each plot, the grey stripe embraces the concentration range in the reference "healthy" population. Colour coding: T1 (red); T2 (blue); COVID-19-R (yellow). * indicates p-value < 0.05: the upper line indicates the statistical significance between all the COVID-19 subjects and the COVID-19-R group; the lower line indicates statistical significance between T1 and T2.
(TIF)

**S11 Fig. Comparison between severe and fatal states through genome-scale metabolic modelling.** Heatmaps showing the clustering of the different organs according to the activity of each reaction (left severe, right fatal state). Each column represents a reaction in the human metabolic network and its color accounts for the (normalized) activity of that specific reaction in the corresponding organ. In the heatmap on the left, three main clusters are labelled with a red, green and orange dot and the organs belonging to each cluster are labelled accordingly and this color code is maintained in the heatmap on the right. Organs changing clusters in the

two heatmaps are connected with a line.
(TIF)

**S12 Fig. Genome-scale metabolic modelling of female subjects.** A) Clustering of context-specific models. Each column represents a reaction in the human metabolic network and its color accounts for the (normalized) activity of that specific reaction in the corresponding context-specific model. Models are clustered according to their profile of active/inactive reactions. B) Heatmap that includes those metabolic processes (subsystems) showing an overall flux increase that paralleled that of disease severity (from mild to severe). Their activity in the healthy-model is also included in the figure for clarity. C) Fold change of average metabolic activity for each subsystem in the healthy vs. fatal condition. D-E) Heatmaps showing the clustering of the different organs according to the activity of each reaction (D- healthy state, E-fatal state). As in in the panel A, each column represents a reaction in the human metabolic network and its color accounts for the (normalized) activity of that specific reaction in the corresponding organ.
(TIF)

## Acknowledgments

P.T., V.G., V.P. and C.L. acknowledge their participation to the NMR International COVID-19 Research Network and the support and the use of resources of Instruct-ERIC, a Landmark ESFRI project, and specifically the CERM/CIRMMP Italy Centre.

## Author Contributions

**Conceptualization:** Paola Turano.

**Formal analysis:** Veronica Ghini, Walter Vieri, Valentina Pecchioli, Tania Alonso-Vásquez.

**Funding acquisition:** Giancarlo Landini, Paola Turano.

**Investigation:** Veronica Ghini, Tommaso Celli, Valentina Pecchioli.

**Methodology:** Veronica Ghini, Marco Fondi, Claudio Luchinat, Paola Turano.

**Resources:** Tommaso Celli, Nunzia Boccia, Lorenzo Pelagatti, Laura Bertini, Vieri Vannucchi.

**Software:** Marco Fondi.

**Supervision:** Marco Fondi, Claudio Luchinat, Laura Bertini, Vieri Vannucchi, Giancarlo Landini, Paola Turano.

**Writing – original draft:** Veronica Ghini, Marco Fondi, Paola Turano.

**Writing – review & editing:** Walter Vieri, Claudio Luchinat.

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
