## [Decision Letter · Decision Letter 0]

6 Sep 2023

Dear Dr. Turano:

Thank you for submitting your manuscript " COVID-19: a complex disease with a unique metabolic signature" for review by PLOS Pathogens. Your manuscript has been evaluated at editorial level and external peer reviewers. The reviewers appreciated the attention to important topic but identified some aspects of the manuscript that should be improved. We therefore request you to modify the manuscript according the review comments before we can consider for acceptance. Your revisions should address these specific points raised by each reviewer.

Sincerely,

Hung Nguyen

Guest Editor

PLOS Pathogens

Sonja Best

Section Editor

PLOS Pathogens

Kasturi Haldar

Editor-in-Chief

PLOS Pathogens

orcid.org/0000-0001-5065-158X

Michael Malim

Editor-in-Chief

PLOS Pathogens

orcid.org/0000-0002-7699-2064

Dear Dr. Turano:

Thank you for submitting your manuscript " COVID-19: a complex disease with a unique metabolic signature" for review by PLOS Pathogens. Your manuscript has been evaluated at editorial level and external peer reviewers. The reviewers appreciated the attention to important topic but identified some aspects of the manuscript that should be improved/ We therefore request you to modify the manuscript according the review comments before we can consider for acceptance. Your revisions should address these specific points raised by each reviewer.

Reviewer Comments (if any, and for reference):

Reviewer's Responses to Questions

**Part I - Summary**

Reviewer #1: In this manuscript, Ghini et. al. investigate the metabolic signature of SARS-CoV-2 infection, in relation to viral variants, vaccination status, sex differences and comorbidities and try to identify metabolites as prognostic markers to the fatal outcome of the disease. Finally, by utilizing a multi-organ metabolic model they attempt to simulate the impact of COVID-19 on the entire host metabolism. The authors try to shed more light on the effects of the SARS-CoV-2 infection on host metabolism and improve our understanding of the pathology of COVID-19. The fact that the authors collected samples from a big cohort and through a broad time period strengthens the study. On the other hand, by employing NMR analysis instead of MS they limit the detection of metabolites and are not able to address all the changes in host metabolism. Overall, the concept of the study is interesting as it addresses understudied aspects of the SARS-CoV-2 infection such as viral variants and vaccination effect. However, there are there are some issues on this study that need to be addressed, in order to improve the quality of the manuscript.

Reviewer #2: The manuscript "COVID-19: a complex disease with a unique metabolic signature" by Ghini et al provide novel dataset describing metabolite and lipoprotein signature in realtion to COVID-19 infection. This is a follow up manuscript to previous work published by Authors. The number of samples used for analysis are good and Authors have provided different comparisons like Sex differences, vaccination, etc. Overall this manuscript provides novel information to the field.

**Part II – Major Issues: Key Experiments Required for Acceptance**

Reviewer #1: 1) In the manuscript it is stated that the samples were obtained at the peak of infection, but it is not clear whether the patients were hospitalized before or after the sampling. Hospitalization and maybe nutrient supplementation, especially for the severe patients and those with fatal outcome, could affect the metabolic signature and in particular the glucose levels. The authors should clarify that the patients received no nutrient supplementation prior to obtaining samples.

2) In figure 8, the authors compare the metabolic activity of fatal patients to healthy controls. Wouldn’t a comparison between fatal and severe (recovered) patients would also give us more indications on which metabolic pathways or organs are more perturbed and could lead to death because of SARS-CoV-2 infection?

Reviewer #2: This work is novel but is very similar to what authors have done previously with ~350 samples. The analysis done are different but the techniques used is similar. So, in future authors should make an effort to make this work comprehensive and with more samples and long duration follow up of the patients if possible.

**Part III – Minor Issues: Editorial and Data Presentation Modifications**

Reviewer #1: 1) In the text for figures 1 and 2 the authors state that there are 11/25 metabolites and 16/30 lipoproteins significantly different compared to the control group. But in the figures, the highlighted number of significantly altered metabolites and lipoproteins is different (for example for variant o there are 16 metabolites and 18 lipoproteins highlighted). Why is that? Are the authors only referring to the metabolites changed in all variant groups? That should be made clear.

2) In figure 1, Ornithine is only changed in the o variant group. In the study by Li et. al. (Frontiers in Immunology, 2021) ornithine was found upregulated and correlating positively with inflammation in severe COVID-19 patients. The authors should comment on that, also regarding maybe the less severity of the o variant compared to previous variants of SARS-CoV-2?

3) On the same context, all the ketone bodies are also more increased in the o variant group compared to other groups. The authors should also comment on that.

4) In figure 3, although the colors are explained in the figure legend, it would be helpful to have a color-coding legend in the figure for VAX and NO-VAX groups.

5) In line 282 it is written that “This analysis revealed a clear cut between…” Something is missing.

Reviewer #2: Please confirm and update that samples used were >500 and not >600 as the author summary section says >600.

PLOS authors have the option to publish the peer review history of their article (what does this mean?). If published, this will include your full peer review and any attached files.

Reviewer #1: **Yes: **Fotios Karagiannis

Reviewer #2: No

Figure Files:

Data Requirements:

Reproducibility:

References:

---

## [Decision Letter · Decision Letter 1]

30 Oct 2023

Dear Prof. Turano,

We are pleased to inform you that your manuscript 'COVID-19: a complex disease with a unique metabolic signature' has been provisionally accepted for publication in PLOS Pathogens.

Best regards,

Hung Nguyen

Guest Editor

PLOS Pathogens

Sonja Best

Section Editor

PLOS Pathogens

Kasturi Haldar

Editor-in-Chief

PLOS Pathogens

orcid.org/0000-0001-5065-158X

Michael Malim

Editor-in-Chief

PLOS Pathogens

orcid.org/0000-0002-7699-2064

Reviewer Comments (if any, and for reference):

Reviewer's Responses to Questions

**Part I - Summary**

Reviewer #1: The authors have addressed adequately all the comments raised by the reviewers, therefore improving the quality of the manuscript.

Reviewer #2: (No Response)

**Part II – Major Issues: Key Experiments Required for Acceptance**

Reviewer #1: (No Response)

Reviewer #2: Thanks to the Author for appropriately addressing the concerns.

**Part III – Minor Issues: Editorial and Data Presentation Modifications**

Reviewer #1: (No Response)

Reviewer #2: (No Response)

PLOS authors have the option to publish the peer review history of their article (what does this mean?). If published, this will include your full peer review and any attached files.

Reviewer #1: **Yes: **Fotios Karagiannis

Reviewer #2: No

---

## [Editor Report · Acceptance letter]

4 Nov 2023

Dear Prof. Turano,

We are delighted to inform you that your manuscript, "COVID-19: a complex disease with a unique metabolic signature," has been formally accepted for publication in PLOS Pathogens.

Best regards,

Kasturi Haldar

Editor-in-Chief

PLOS Pathogens

orcid.org/0000-0001-5065-158X

Michael Malim

Editor-in-Chief

PLOS Pathogens

orcid.org/0000-0002-7699-2064